# The effect of growth rate on otolith-based discrimination of cod (*Gadus morhua*) ecotypes

Einar Pétur Jónsson[1,2]*, Steven E. Campana[1ʘ], Jón Sólmundsson[2ʘ], Klara B. Jakobsdóttir[2‡], Hlynur Bárðarson[2‡]

**1** School of Engineering and Natural Sciences, University of Iceland, Askja, Reykjavík, Iceland, **2** Marine and Freshwater Research Institute, Hafnarfjörður, Iceland

ʘ These authors contributed equally to this work.
‡ These authors also contributed equally to this work
* einarpeturjonsson@gmail.com

**Data Availability Statement:** All relevant data are within the manuscript and its Supporting Information files.

## Abstract

Otolith shape has previously been used to identify ecotypes within the Icelandic cod (*Gadus morhua*) stock, using DST profiles to validate the results. Fish otolith shape variation has repeatedly been found to be largely determined by growth rate. To examine the effect of growth rate on the relationship between otolith shape and cod ecotypes (using the *Pan* I genotype as a proxy for ecotype), 826 archived sagittal otoliths collected over a 58 year sampling period were retrieved, the individual growth rate calculated, and otolith shape described using both Normalized Elliptic Fourier transform and Discrete Wavelet transform. Discriminant functions of otolith shape successfully classified ecotype, whether using Fourier or Wavelet descriptors, but only when excluding a heterozygous genotype from the analysis. The otolith shape variability of this genotype lowered the classification success, while otolith shape, in turn, was significantly affected by growth rate and cohort. Growth rate differences previously reported for the ecotypes were present, but were less marked than expected and indeed, growth rate variance attributable to ecotype identity was dwarfed by cohort- and location-related variance in growth. Such a strong effect of growth rate suggests that cod ecotype discrimination based on otolith shape is sensitive to both temporal and spatial variations in growth, which can mask the effect of ecotype-related growth rate differences on otolith shape.

## Introduction

Ecotypes are sub-groups of populations differing in allele frequency across loci, formed through their multiple trait adaptations to environmental variables [1]. The ecological separation that is entailed in the formation of ecotypes has been linked to niche partitioning, where a single species may occupy a wider environmental gradient through a number of distinct ecotypes within the species, as exemplified by the cyanobacterium *Prochlorococcus* [2] and lake whitefish (*Coregonus clupeaformis*) [3]. The ecotype stage has been suggested to be a precursor

**Funding:** The authors reveived no specific funding for this work.

**Competing interests:** The authors have declared that no competing interests exist.

to evolutionary processes such as speciation, and this divergence has been studied extensively [1] since Turesson's studies on saltbushes in Sweden, where he coined the ecotype term [4]. Ecotypes and the divergence they entail are now common subjects of study in fish such as Arctic char (*Salvelinus alpinus*) [5] and Atlantic cod [6]. Increased attention is also being given to maintaining the diversity that ecotypes confer to species and ecosystems, especially in the context of sustainable fisheries management [7,8].

Two distinct ecotypes of Atlantic cod (*Gadus morhua*) have been repeatedly reported from throughout the trans-oceanic range of the species [9–11]. Off the coast of Iceland, data storage tag (DST) profiles have been used to identify a coastal ecotype and a more migratory (frontal) ecotype associated with thermal fronts [6,12,13]. In addition, there is evidence of fine-scale differences in spawning habitat selection between the ecotypes [6]. The ecotypes present morphological differences [14,15], as well as genetic divergence at the pantophysin I locus (*Pan* I). The *Pan* I$^{AA}$ genotype is strongly associated with the coastal ecotype and the *Pan* I$^{BB}$ genotype with the frontal ecotype, while the *Pan* I$^{AB}$ displays both types of behaviour [14,16,17]. Linking the *Pan* I genotypes with DST profiles has shown 94% of *Pan* I$^{AA}$ individuals present coastal behaviour and 88% of the *Pan* I$^{BB}$ indivudals present frontal behaviour [11]. Thus, the frequency of the A allele decreases by 0.44% per depth meter when analysing catch data [17]. Although there is unarguably a level of error involved, it is reasonable to use the *Pantophysin* I locus as a convenient proxy for the ecotype identity.

Discrimination of stock components is important in fisheries science, and can be based on a broad range of methods [18,19]. Otoliths are metabolically inert calcified structures traditionally used in age reading, but having shown great potential in stock discrimination [20]. Although the otolith's shape is unique to each species, both internal and external otolith shape have been used to discriminate between stock components [18,20,21]. The external otolith shape has recently been used to predict the ecotype identity of cod around Iceland, with DST profiles and *Pan* I genotyping used to validate the results [12].

Otolith shape variation has been associated with both intrinsic and extrinsic factors, but some of these factors have been suggested to be ultimately acting on otolith shape through the intermediary of growth variability among individuals. Both genetic and environmental determinants of otolith shape have been detected, suggesting dual shape regulation [21,22]. Despite the influence of environmental factors such as diet [23] and temperature [24] on otolith shape, the strongest link appears to be that between otolith shape and fish growth rate [18]. Indeed, environmental factors affecting otolith shape have also been proposed to be acting via individual growth rate [21,25]. The cause of otolith shape variation in cod ecotypes has not been confirmed, but otolith shape differences are driven by growth rate in other cod stocks [18]. As the coastal ecotype has been found to grow faster than its frontal counterpart [9,26,27], but see [14], it is reasonable to assume that such a relationship between otolith shape and growth rate exists here as well.

The objective of this study was to investigate the effect of growth rate on the relationship between otolith shape and cod ecotypes off Iceland. The study examined cod from three separate spawning grounds collected over a period of half a century. Archived otoliths were retrieved, and their shape described using both Normalized Elliptic Fourier transform and Discrete Wavelet transform. The otoliths had previously already been genotyped for the *Pan* I locus by Jakobsdóttir *et al.* [27], and this genetic information was used as a proxy for ecotype identity in the current study. After using otolith shape to distinguish among ecotypes, the relative contributions of cohort, location, ecotype identity and growth rate to the shape variation were evaluated. We hypothesized that ecotype discrimination based on otolith shape would be particularly sensitive to spatial and temporal variations in fish growth rate.

## Methods

### Sampling and measurements

Samples of Icelandic cod were collected both at port and on surveys by the Icelandic Marine Research Institute (MRI) using three types of commercial fishing gear between 1948 and 2000 (Table 1). Most samples were collected using gill nets and as no year-location combination had more than one type of gear, gear selectivity could not be evaluated. All gear types did however catch fully recruited sizes of either mature or spawning cod. Cod were caught from late March through April (season of spawning) at three known spawning grounds [28] west and southwest of Iceland: Breiðafjörður (BRE), Faxaflói (FAX) and the southwest (SW) (Fig 1). Samples were collected over many decades, in support of various studies, resulting in an unavoidable unbalanced design for the current analysis. Use of this unique archive offered (1) an extended time series that could be used to identify short term anomalies and provide the necessary contrast in growth rate to study its effect on otolith shape discrimination, and (2) genetic information for the *Pan* I locus, thus providing an ecotype proxy over a long time scale. The locations of three samples in the southwest area (years 1959, 1966 and 1985) are approximate.

The total length of each cod was measured and their stage of sexual maturity determined. Sagittal otoliths were removed from the fish and their age determined by experienced otolith age-readers. The otolith growth increments were counted from the core to the edge of the otolith in a transverse section through the otolith core, perpendicular to the otolith's longest axis. The age range was from 6 to 14 years.

Otolith shape analysis was based on the intact (unaged) sagittal otolith of each pair, which was photographed and measured following the methods detailed by Bardarson *et al*. [12]. The right sagittal otolith, with the *sulcus acusticus* facing up and the dorsal side pointing towards the top of the image, was photographed under a Leica MZ6 stereomicroscope at 0.63x magnification using a Plan Apo 0.4x objective. In cases where the right otolith had been sectioned for ageing (1/4 of all samples), an image of the left otolith was flipped horizontally and used instead. Strong backlighting of the otolith resulted in a contrasting dark shape on a light

**Table 1. Overview of the *Pan* I genetically identified cod (*G. morhua*) samples collected between 1948 and 2000 originally described in Jakobsdóttir et al. (2011) [27].**

| Sampling year | n | Location | Depth (m) | Gear type | AA | AB | BB | Nr. of cohorts |
|---|---|---|---|---|---|---|---|---|
| 1948 | 68 | BRE | 100 | Longline | 16% | 61% | 26% | 9 |
| 1957 | 84 | SW | 37 | Gill net | 15% | 70% | 15% | 8 |
| 1959 | 61 | SW | 37–74* | Gill net | 11% | 47% | 45% | 7 |
| 1966 | 67 | SW | 37–74* | Bottom trawl | 4% | 87% | 9% | 7 |
| 1972 | 69 | FAX | 38–83* | Gill net | 4% | 70% | 26% | 6 |
| 1973 | 73 | SW | 70 | Gill net | 3% | 81% | 16% | 6 |
| 1976 | 52 | FAX | 55 | Gill net | 35% | 65% | 0% | 6 |
| 1976 | 47 | SW | 45 | Bottom trawl | 70% | 28% | 2% | 6 |
| 1979 | 61 | FAX | 50 | Gill net | 0% | 98% | 2% | 5 |
| 1985 | 73 | SW | 37–74* | Gill net | 19% | 55% | 26% | 8 |
| 1996 | 40 | BRE | 63 | Gill net | 28% | 72% | 0% | 5 |
| 1996 | 25 | FAX | 55 | Gill net | 92% | 8% | 0% | 5 |
| 1996 | 68 | SW | 56 | Gill net | 57% | 43% | 0% | 8 |
| 2000 | 38 | BRE | 150 | Gill net | 21% | 66% | 13% | 5 |

Depth distributions (10–90% quantiles) within location are presented where the exact depth of catch was unavailable (entries with asterisks). These samples were analyzed for their otolith shape and their growth rate index (G).

background. The images were then converted to 8-bit binary images, creating a black shape on a white background, followed by measurement of the otolith's maximum Feret diameter (the maximum distance between any two points on the otolith's edge). All image manipulation and measurements were carried out with the Fiji software [29]. As 16% of the otoliths were broken, these were glued together to maintain sample size, following Bardarson's method and analyses supporting such otolith reconstruction [30].

The shape of the otolith outline was analysed using the ShapeR package [31] for the R software [32]. In this procedure, the dark otolith image was distinguished from the white background and the otolith's outline detected. Two independent methods were used to generate shape descriptors from the otolith outline: (1) Normalized Elliptic Fourier transform generated 45 Normalized Fourier coefficients that describe and quantify the shape with sines and cosines [31]. The Fourier coefficients were normalized with regards to otolith length (maximum Feret diameter), as part of the Normalized Elliptic Fourier analysis. (2) Discrete Wavelet transform was better suited for the description of localized changes in shape, such as sharp edges, as the 64 Wavelet descriptors on 10 levels detect shape variations at specific regions of the otolith [31,33]. The accuracy of the otolith outline reconstruction can be seen in supplementary material S1 Fig.

All shape descriptors were used in the analyses. In general, the accuracy of shape description improves as the number of shape descriptors is increased, although the first 10 Fourier coefficients commonly describe most of the shape [12,18].

The otolith shape descriptors were detrended to remove any correlation with fish length. For each shape descriptor significantly affected by fish length, a general linear model (GLM) was fitted with the shape descriptor as a function of fish length and location. The shape descriptor was then detrended by subtracting the product of the fish length and the within-group slope (β) acquired from the GLM. The resulting detrended shape descriptor was thus comparable among fish of different lengths from different locations.

The *Pan* I locus genotype was used as the exclusive indicator for ecotype identity, as the AA genotype is associated with the coastal ecotype, the BB genotype is associated with the frontal ecotype, and the heterozygote AB displays both frontal and coastal behaviour [12]. This genetic information was obtained from tissue remaining on the otoliths and has previously been reported by Jakobsdóttir *et al.* [27].

## Data analysis

To analyse all age groups simultaneously and still retain a relative measure of fish growth rate, a growth rate index was created. This was done by first fitting a re-parameterized von Bertalanffy growth equation [34] to length-at-age data using nonlinear least squares (Fig 2).

$$E[L|t] = L_1 + (L_3 - L_1) * \frac{1 - r^{2*\frac{t-t_1}{t_3-t_1}}}{1 - r^2}$$

where

$$r = \frac{L_3 - L_2}{L_2 - L_1}$$

**Eq 1.** The von Bertalanffy growth function as re-parametrized by Francis (1988).
$L_1$, $L_2$ and $L_3$ represent length at ages $t_1$, $t_2$ and $t_3$, respectively.

The von Bertalanffy function was fitted to the data using the function *vbfr* from the fish-methods package [35] for R. A growth rate index (*G*) was defined as the residual (*res$_i$*) for a given individual fish *i* at age *j* from the length predicted by the von Bertalanffy growth function

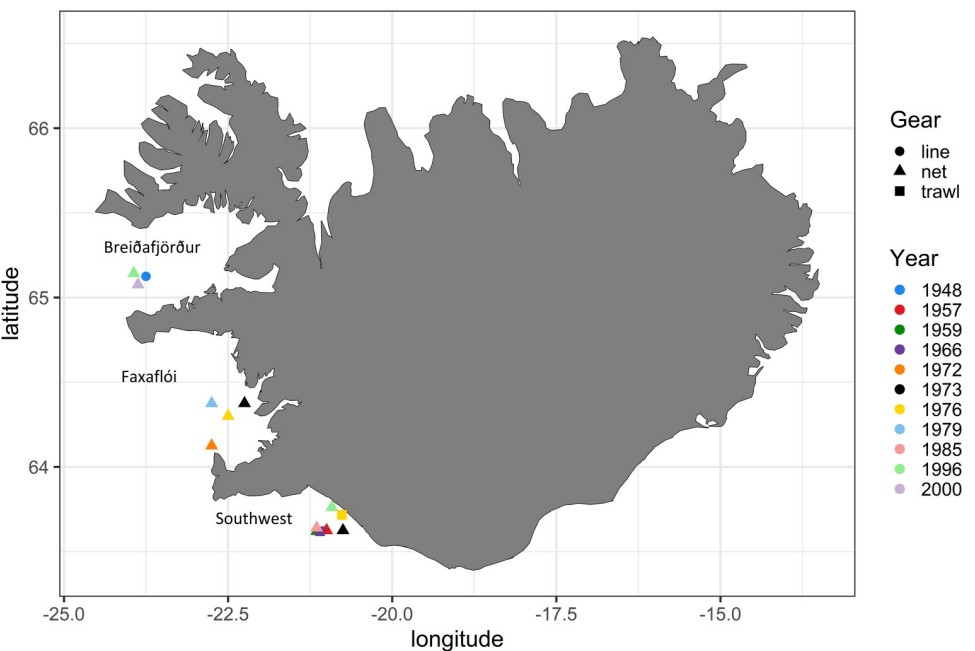

**Fig 1. Map of the sampling locations.** Colour indicates sampling year and shape indicates gear type.

at that age ($\hat{y}_j$). Thus, the magnitude of *G* indicates how much smaller or larger an individual fish is relative to the predicted length, and therefore allows comparison of relative growth rates across age groups.

Discriminant function analysis (DFA) was used to identify the shape descriptors most effective in discriminating among the *Pan* I groups. Unbiased classification accuracy was assessed using cross validation where the prior probabilities were computed from the group sizes.

Multivariate General Linear Models were fitted to the data to test for effects of growth rate, location, cohort and *Pan* I identity on the shape descriptors. Analyses of Variance (ANOVAs) were used to evaluate the variance of growth rate among *Pan* I genotypes, locations and cohorts. A Shapiro-Wilks test indicated some non-normality of the distribution of residuals from the ANOVAs, but ANOVAs tend to be robust to modest deviations from normality [36–38].

SPSS version 26 [39], the R software (version 3.6.1) [32] and the tidyverse package [40] were used for statistical analyses and graphics presented in the study. The alpha-level threshold for significance was 0.05 for all analyses.

## Results

Of the three *Pan* I genotypes, AB was generally the most numerous, accounting for a mean proportion of 0.60 across cohorts and locations, while the mean proportion of AA was 0.24 and that of BB 0.16. There were clear changes in *Pan* I proportions through time, although not among locations (Fig 3). There were significant differences in *Pan* I proportions across cohorts and age groups (Chi-squared tests, $p<0.01$, n = 826). The most prominent cohort-related variations were the inversions in the proportions of the AA and AB genotypes. However, these cohort-specific *Pan* I proportions did not vary significantly between the three locations (Chi-squared test, $p>0.05$, n = 826).

Otolith shape analysis identified the three *Pan* I genotypes with relatively low accuracy. Stepwise Discriminant function analyses (DFAs) of *Pan* I genotypes using either Fourier

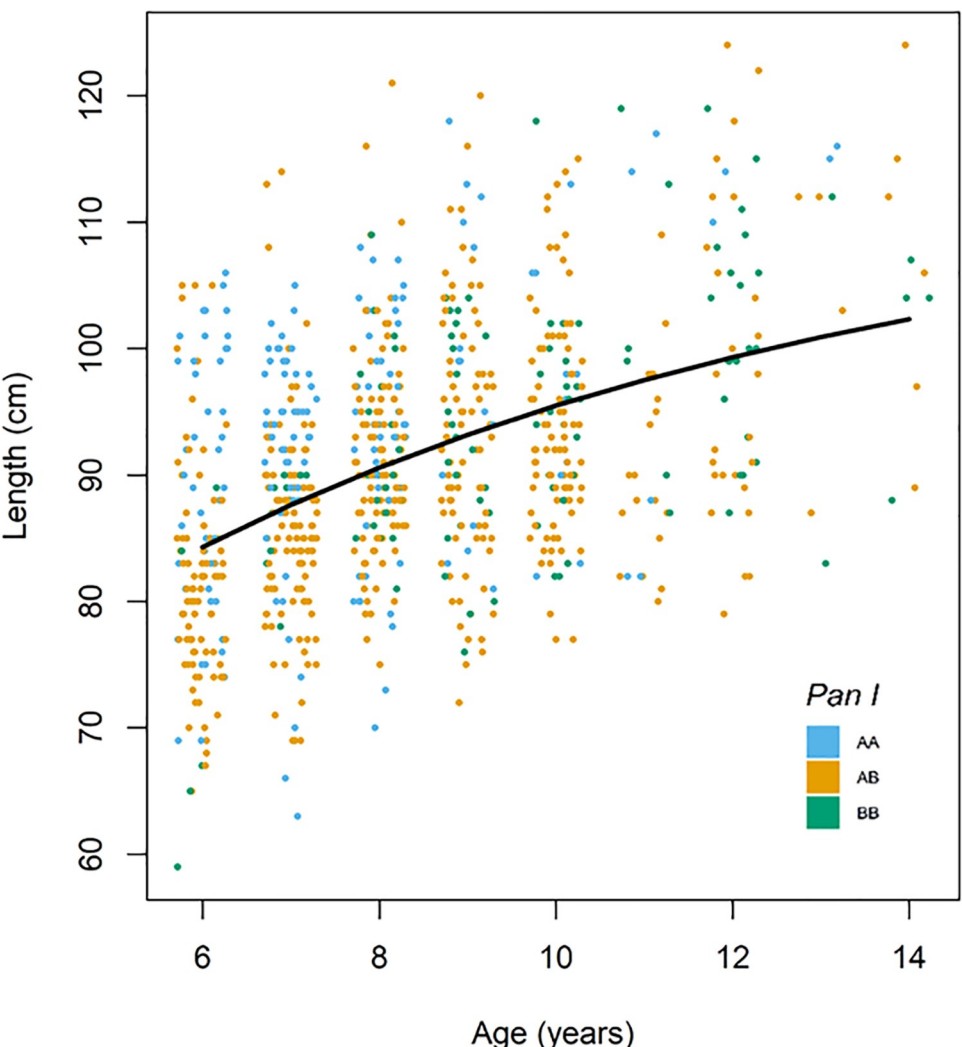

**Fig 2. A re-parameterized von Bertalanffy growth curve fitted to the length-at-age data.** Each dot represents an individual and its residual from the curve is the individual's relative growth rate ($G$). N = 826. Data are jittered within age-groups.

coefficients or Wavelet descriptors resulted in two significant discriminant functions incorporating up to 10 Fourier coefficients in one analysis and 15 Wavelet descriptors in the other. Cohort and $G$ were selected in both DFAs, while location was not selected at any step in either analysis ($p<0.01$, n = 826). Classification success using these discriminant functions was 68%, which was similar to DFA functions without cohort and $G$. The classification success was in all cases highest for AB (92–95%), the most numerous of the *Pan* I genotypes, while AA and BB were classified with 17–39% and 2–5% success, respectively (Table 2). AA and BB were most commonly (60–90%) misclassified as AB.

Discriminant function analyses of only the *Pan* I^AA and *Pan* I^BB genotypes resulted in markedly higher classification success than the DFAs of all three genotypes. Stepwise DFAs incorporated 10 Fourier and 9 Wavelet descriptors (as well as cohort) and classified the *Pan* I^AA and *Pan* I^BB genotypes with over 85% and 72% success, respectively (Table 3). Although the discriminant scores of the *Pan* I^AA and *Pan* I^BB genotypes were reasonably well separated,

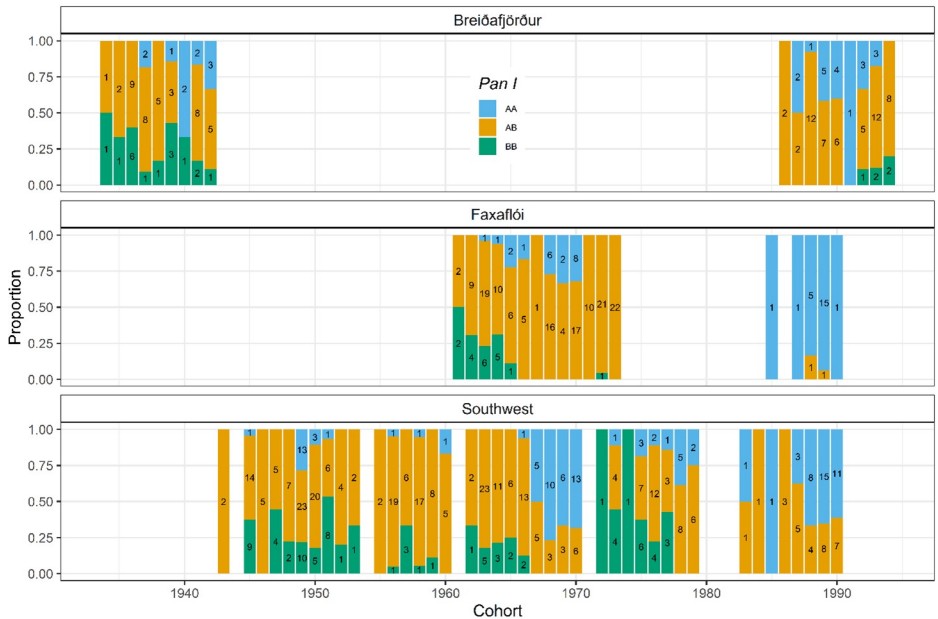

**Fig 3. Relative proportions of the three cod (*G. morhua*) *Pan* I genotypes across cohorts in the three sampling locations: Breiðafjörður (n = 146), Faxaflói (n = 207) and Southwest (n = 473).** Sample sizes for the genotypes within cohorts are indicated within bars.

the discriminant scores of the AB genotype were intermediate in value, and distributed across the entire range of AA and BB scores (Fig 4A–4F). When classified with the *Pan* I$^{AA}$ or *Pan* I$^{BB}$ discriminant functions, AB fish were split relatively evenly between the AA and BB genotypes (Table 4).

Although the *Pan* I genotype could be weakly predicted by otolith shape, otolith shape in turn was significantly influenced by the growth rate of the fish. Multivariate GLMs indicated that *G* significantly affected 18 of the 42 Fourier descriptors and 16 of the 63 Wavelets. Adding location, cohort and the *Pan* I identity to the analysis revealed that neither location nor *Pan* I identity had a significant effect on shape, while both *G* and cohort had a significant effect on these shape descriptors, with cohort and its interaction with *G* contributing most to the model ($p < 0.01$, n = 826 for all analyses).

The growth rate of the three genotypes differed at a gross scale, but these differences disappeared once account was taken of growth rate differences across location and cohort. A one-way Analysis of Variance (ANOVA) revealed statistically significant differences in *G* among the *Pan* I genotypes when all data were pooled ($F(2, 826)$, $p < 0.01$, Fig 5). A post-hoc Tukey's

**Table 2. Classification success (%) of the cod (*G. morhua*) *Pan* I genotypes.**

| Model | *Pan* I$^{AA}$ | *Pan* I$^{AB}$ | *Pan* I$^{BB}$ |
|---|---|---|---|
| *Pan* I~ Wavelet | 17.0 | 94.5 | 5.0 |
| *Pan* I~ Fourier | 29.1 | 92.6 | 5.0 |
| *Pan* I~ Wavelet + cohort + G | 34.1 | 93.1 | 2.5 |
| *Pan* I~ Fourier + cohort + G | 39.0 | 91.8 | 1.7 |

Discriminant function analyses of otolith shape showing overall classification success between 64 and 68% for all analyses. N = 826 for each analysis.

Table 3. Classification success (%) of the cod (*G. morhua*) *Pan* I$^{AA}$ and *Pan* I$^{BB}$ genotypes.

| Model | *Pan* I$^{AA}$ | *Pan* I$^{BB}$ |
|---|---|---|
| *Pan* I~ Wavelet | 86.3 | 63 |
| *Pan* I~ Fourier | 81.9 | 72.3 |
| *Pan* I~ Wavelet + cohort | 87.9 | 72.3 |
| *Pan* I~ Fourier + cohort | 85.2 | 74.8 |

Discriminant function analyses of otolith shape discriminating only between the *Pan* I$^{AA}$ and *Pan* I$^{BB}$ genotypes.
N = 301 for each analysis.

test indicated that *G* for the *Pan* I$^{AA}$ genotype had a significantly higher growth rate than the AB and BB genotypes ($p < 0.05$) and that the latter two did not differ between themselves. However, when accounting for location and cohort, there were significant variations in *G* among locations ($F_{(2, 826)}$, $p < 0.001$) and cohorts ($F_{(55, 826)}$, $p < 0.001$, Fig 6), but not among the *Pan* I genotypes ($F_{(2, 826)}$, $p = 0.638$) (3-way ANOVA, see Table 5A). The only significant interaction in the analysis was that between location and cohort ($F_{(19, 826)}$, $p < 0.01$). Excluding the *Pan* I$^{AB}$ from the analysis gave similar results, except that the interaction between *Pan* I and location was significant, while the location-cohort interaction term was not (Table 5B). Growth rate was consistently lowest in Breiðafjörður, both with the *Pan* I$^{AB}$ genotype (Tukey's test, $F_{(2, 826)}$, $p < 0.001$) and without (Tukey's test $F_{(2, 301)}$, $p < 0.001$).

Differences in fish growth rate between genotypes were not consistent when accounting for location. One-way ANOVAs revealed significant differences in *G* between the *Pan* I genotypes in both Faxaflói ($F_{(2, 207)}$, $p < 0.01$) and the Southwest ($F_{(2, 473)}$, $p < 0.01$), and post-hoc Tukey's tests showed this to be the AA genotype separating from the other two genotypes in both cases ($p < 0.05$, see Fig 5). However, no significant differences in *G* were found between the genotypes in Breiðafjörður ($F_{(2, 146)}$, $p = 0.9$). When accounting for cohort as well as location, no consistent differences could be identified in fish growth rate between the genotypes in any of the three locations (see Fig 7 for example).

## Discussion

There were clear cohort-related variations in genotype proportions in all three locations, with inversions in genotype dominance occurring between cohorts. The *Pan* I genotypes have been shown to use similar spawning grounds [14], with the separation in depth they inhabit during spawning season being greatly reduced [6,12] so it was to be expected that random sampling of these spawning grounds would yield proportions similar to those in the spawning population. No significant variation in proportions were detected among locations, despite the sampling in Breiðafjörður taking place at greater depth (Table 1) and the frequency of the B allele being positively correlated with depth [15,16]. Overall, the *Pan* I$^{AB}$ genotype was most frequent in all three locations, as has been pointed out in other studies [41], but there were also strong changes in genotype proportions among cohorts, both at shorter and longer time scales. As there was no identifiable trend in genotype proportions within each sampling year, regardless of the cohort (Table 1), we conclude that the cohort-related variations were not due to biased sampling of the genotypes. Assuming these changes in genotype proportions between cohorts were not the product of random variation or small-scale habitat preferences by certain genotypes, then the year-class strength of each genotype seems to vary. Similar temporal and spatial variations in ecotype proportions have also been noted along the Norwegian Skagerrak coast for the Fjord and North Sea ecotypes present there [8].

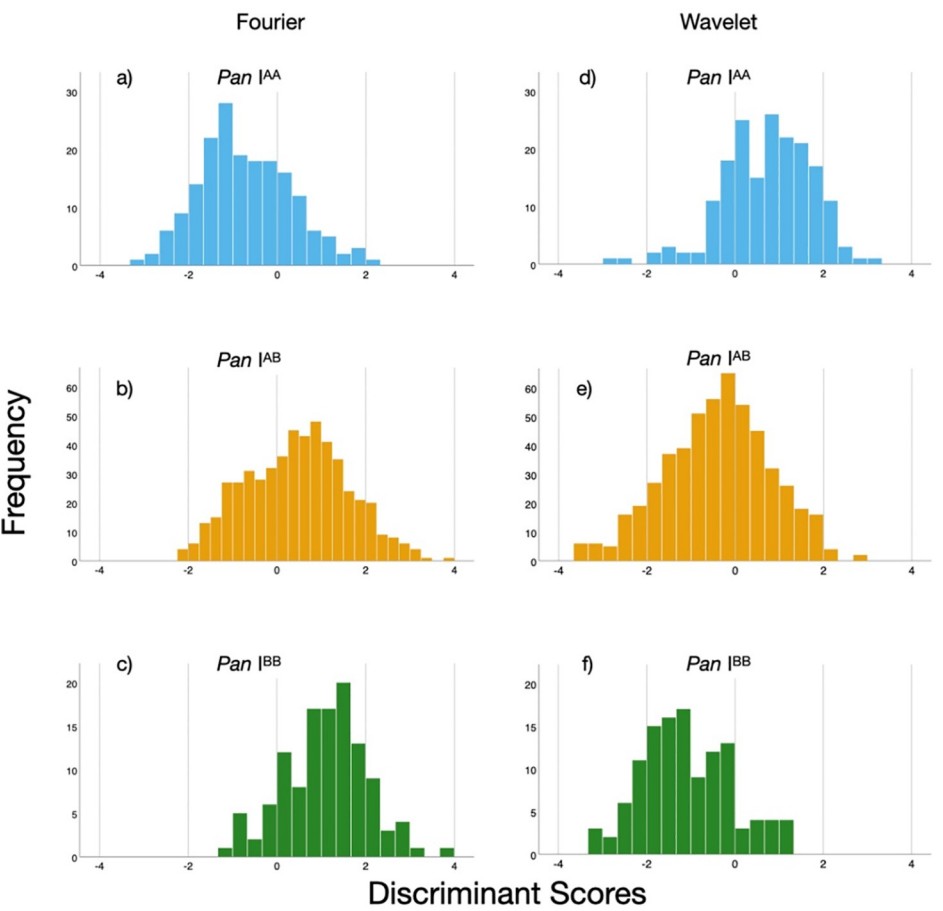

**Fig 4.** a-f. Distributions of the three cod (*G. morhua*) *Pan* I genotypes along the discriminant scores from Discriminant function analyses of the *Pan* I^AA and *Pan* I^BB genotypes using cohort and Fourier (a-c) or Wavelet descriptors (d-f) as discriminants.

Discriminating genotype identity using otolith shape yielded lower success than expected, both when using the Fourier coefficients and Wavelet descriptors. The low average classification success of 68% contrasted with the higher overall classification success of 77% presented by Bardarson *et al*. [12] when classifying behavioural ecotypes using otolith shape. The Bardarson study reported good success classifying the frontal (90%) and coastal (91%) ecotypes, but less success for the intermediate ecotypes in the analysis. In contrast, the classification success in the present study was highest for the *Pan* I^AB genotype; the *Pan* I^AA and *Pan* I^BB genotypes were usually misclassified as the AB genotype in our study. It is therefore apparent that the

**Table 4. Proportion of cod (*G. morhua*) *Pan* I^AB individuals assigned to *Pan* I^AA and *Pan* I^BB genotypes.**

| Model | *Pan* I^AB assigned to *Pan* I^AA (%) | *Pan* I^AB assigned to *Pan* I^BB (%) |
|---|---|---|
| *Pan* I~ Wavelet + cohort | 53 | 47 |
| *Pan* I~ Fourier + cohort | 47 | 53 |

Using the discriminant scores from Discriminant function analyses of the otolith shape of *Pan* I^AA and *Pan* I^BB genotypes, *Pan* I^AB individuals were assigned to either genotype (N = 301 for each DFA and 525 for AB assignation).

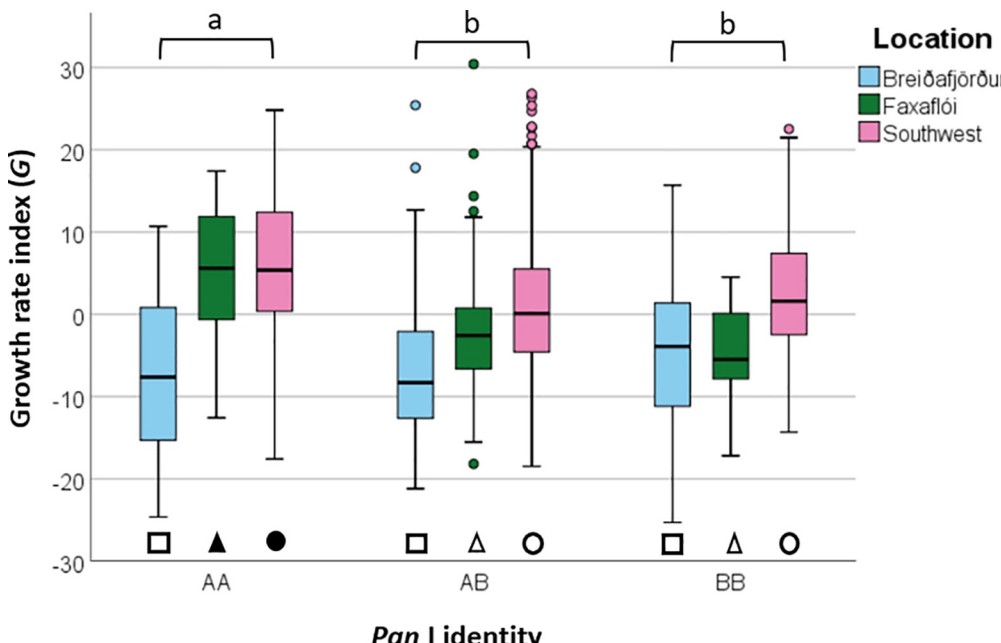

**Fig 5. Growth rate index (*G*) for the cod (*G. morhua*) ecotypes represented by the *Pan* I genotypes.** Letters indicate significant differences among *Pan* I genotypes pooled across locations. Shapes are location-specific, and their fill indicates significant differences among *Pan* I genotypes within each location. Error bars represent 95% confidence interval.

otolith shape differences between the genotypes were not strong enough to allow for successful discrimination. The difficulty in discriminating the heterozygous genotype in the DFAs was demonstrated by excluding this genotype from one of our analyses, which increased the classification success for the *Pan* I[AA] and *Pan* I[BB] genotypes to levels similar to those reported by

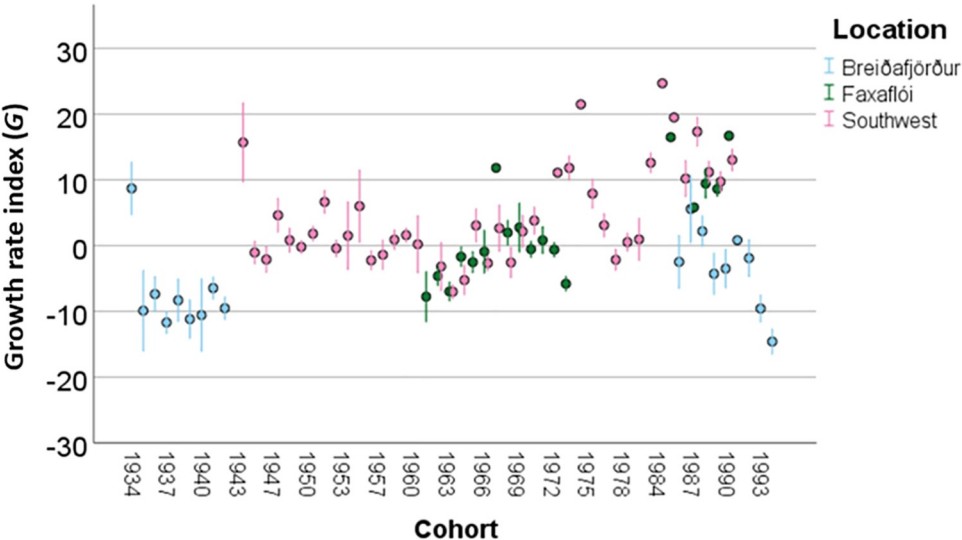

**Fig 6. Variation in the growth rate index (*G*) across cohorts and among locations.** Error bars indicate one standard error.

**Table 5A. Relative effect of location, cohort and *Pan* I identity on the growth rate index (*G*).**

| Source | Type III Sum of Squares | df | Mean Square | F | p |
|---|---|---|---|---|---|
| Corrected Model | 37523 | 146 | 257 | 5.3 | <0.001 |
| Intercept | 3 | 1 | 3.0 | 0.062 | 0.804 |
| *Pan* I | 43 | 2 | 21.9 | 0.45 | 0.638 |
| Location | 1965 | 2 | 982.7 | 20.2 | <0.001 |
| Cohort | 11728 | 55 | 213.3 | 4.4 | <0.001 |
| Location * Cohort | 2195 | 19 | 115.6 | 2.4 | <0.001 |
| *Pan* I * Cohort | 3945 | 64 | 61.6 | 1.3 | 0.085 |
| *Pan* I * Location | 28 | 3 | 9.5 | 0.19 | 0.900 |
| Error | 33037 | 679 | 48.7 | | |
| Total | 70561 | 826 | | | |
| Corrected Total | 70561 | 825 | | | |

Results of a three-way ANOVA evaluating the relative effect of location, cohort and *Pan* I identity on the growth rate index (*G*), as well as their two-way interactions. N = 826. See S1 Table for parameter estimates.

Bardarson *et al.* [12] (Table 3)–an impressive increase, even when considering a part of it may be have been due to the reduction of groups in the analysis. This is further supported by the distribution of the discriminant scores from the two-group DFA, in which the *Pan* I$^{AB}$ genotype was distributed along the entire gradient between the *Pan* I$^{AA}$ and *Pan* I$^{BB}$ endpoints; the otolith shape of *Pan* I$^{AB}$ ranged from a "*Pan* I$^{AA}$ shape" to a "*Pan* I$^{BB}$ shape" (Fig 4 and Table 4). This is consistent with DST data showing the AB genotype to present intermediate behaviour and adaptive plasticity to that of AA and BB ([11] and Supplementary Material S2 Table), thus complicating its discrimination. It seems likely that *Pan* I$^{AB}$ individuals are subject to greater variation in environmental parameters such as temperature and food availability, which in turn affects otolith shape through fish growth [23,25,42]. Furthermore, the large temporal and spatial scale of our study almost certainly includes shape variation independent of the genotype which can affect the classification success for all genotypes.

**Table 5B. Relative effect of location, cohort and *Pan* I identity on the growth rate index (*G*), using only *Pan* I$^{AA}$ and *Pan* I$^{BB}$ genotypes.**

| Source | Type III Sum of Squares | df | Mean Square | F | p |
|---|---|---|---|---|---|
| Corrected Model | 16391 | 88 | 186.3 | 3.4 | <0.001 |
| Intercept | 17 | 1 | 16.6 | 0.3 | 0.583 |
| *Pan* I | 27 | 1 | 26.7 | 0.5 | 0.487 |
| Location | 1168 | 2 | 584.2 | 10.6 | <0.001 |
| Cohort | 6584 | 49 | 134.4 | 2.4 | <0.001 |
| Location * Cohort | 463 | 15 | 30.9 | 0.6 | 0.901 |
| *Pan* I * Cohort | 1512 | 18 | 84.0 | 1.5 | 0.082 |
| *Pan* I * Location[a] | - | - | - | - | - |
| Error | 11637 | 212 | 54.9 | | |
| Total | 29325 | 301 | | | |
| Corrected Total | 28027 | 300 | | | |

Results of a three-way ANOVA evaluating the relative effect of location, cohort and *Pan* I identity on the growth rate index (*G*), as well as their two-way interactions. Only individuals of the *Pan* I$^{AA}$ and *Pan* I$^{BB}$ genotypes were used. N = 301. See S2 Table for parameter estimates.
[a]Insufficient sample size.

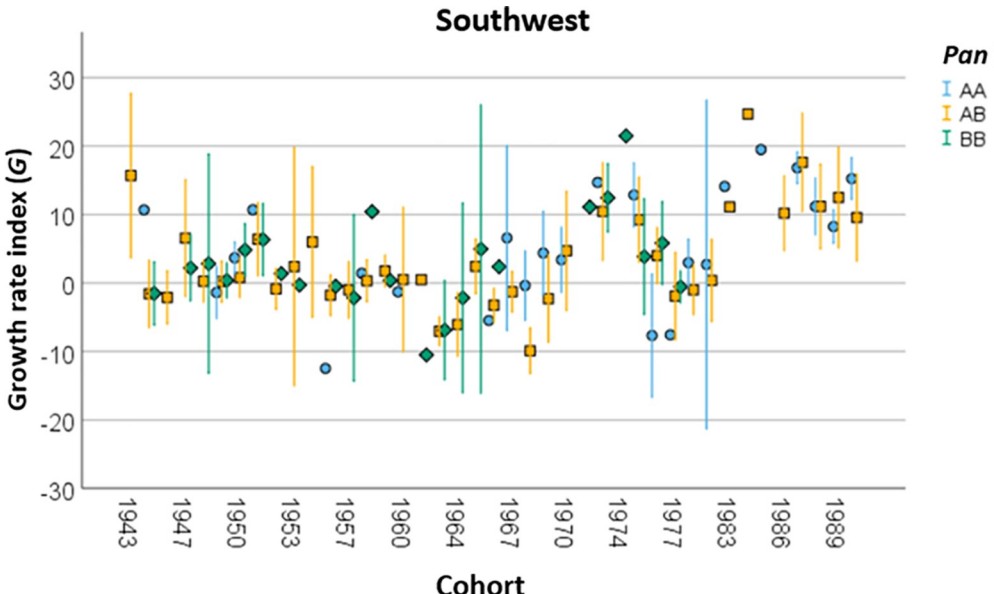

**Fig 7. Growth rate (*G*) of the cod (*G. morhua*) *Pan* I genotypes across cohorts in the Southwest location.** Error bars indicate one standard error.

Both the Fourier and Wavelet descriptions of otolith shape were affected by fish growth. The calculation of an age-independent growth rate index (*G*) indicated that growth rate, as well as cohort, had a significant effect on otolith shape. While otolith shape is determined both by environmental and genetic influences [22], a link between individual growth rate and otolith shape has repeatedly been found [18,43–45] and environmental variables have been suggested to influence otolith shape through its effect on growth rate [21,25]. The relationship between growth and shape presented here is concordant with other studies which have found that fish growth is an important factor to consider in stock discrimination based on otolith shape, as stocks with similar growth rates tend to have similar otolith shapes [18,19]. Thus, it is reasonable to extend that logic to the cod genotypes–that their otolith shapes become more distinguishable with increasing differences in growth rate.

Variation in *G* was greater between locations and among cohorts than between the genotypes, which is not consistent with growth rate differences previously suggested by other studies [9,26,27]. However, spatial and temporal effects have not been considered in previous studies on the growth rate of the cod ecotypes or *Pan* I genotypes, with the exception of one suggestion that the relationship between growth rate and the *Pan* I genotypes was more complicated than previously thought, and that growth rate varied more among spawning areas than among the *Pan* I genotypes within each area [14]. Indeed, when comparing the variance in *G* among locations, cohorts and *Pan* I identity, the genotype identity did not have a significant effect on *G*, while both location and cohort were significant factors (Table 5). Although the spawning grounds are only inhabited by cod during a small part of the year, the fish still retain some spatial separation during the foraging season [46], and are thus exposed to different environments. A recent study on Northeast Arctic cod found environmental factors to exert a joint, indirect influence on otolith shape through changes in fish growth rate [25]. Growth rate variations among locations and cohorts are common in fish and are strongly driven by temperature and food supply [47,48], as could be the case here.

Growth rate's effect on discrimination analyses based on otolith shape has been noted in several species, including haddock (*Melanogrammus aeglefinus*) spawning components on Georges Bank [19], blue whiting (*Micromesistius poutassou*) morphotypes in the Northeast Atlantic [49] and herring (*Clupea harengus*) stock components in the Celtic and Irish seas [50]. Although otolith shape varied more with growth rate and cohort than with ecotype identity in the present study, it appears likely that ecotype-related variations in shape were masked by growth-driven variations in shape across cohorts and location. As cod spawn in a number of areas around Iceland and each ecotype can be found in widely separated spawning grounds, individuals of the same ecotype are likely exposed to different environments, which can then influence their growth [14]. Thus, discrimination of ecotypes based on shape should take account of spatial scale, as the discrimination of individuals inhabiting different environments will be confounded by the variation in growth rate.

Use of the *Pan* I locus as an ecotype proxy understandably comes with a degree of error. Although a strong association between the locus and the behavioural ecotypes has been detected [11,16,17], the relationship is not absolute. In comparison of a DST profile classification with *Pan* I information, a clear gradient across four behavioural ecotypes (coastal, frontal and two intermediate groups) has been found, with coastal individuals genotyped as *Pan* I$^{BB}$ making up only 15% of the total and the frontal ecotype (classified as *Pan* I$^{AA}$) only 16% (according to H. Bardarson (written communication, April 2020)). Using the locus as an ecotype proxy is very convenient in cases where behavioural information is not available, where its use permits approximate ecotype assignment over a long study period when no other ecotype approximation is at hand, such as studies based on archived samples.

Cod ecotypes have been reported elsewhere, with the Norwegian coastal and Northeast Arctic cod ecotypes being the most extensively studied. These ecotypes are genetically distinct [8] and are readily distinguished using either internal or external otolith morphology [51]. Although such discrimination based on otoliths has not been successfully carried out on resident and migratory cod ecotypes in the Northwest Atlantic, cod ecotypes differing in body shape and coloration are readily distinguished in the Gulf of Maine [52] and genetic differentiation between spawning groups has been detected in the area [53]. Although not yet described as ecotypes, genetic differences indicating strong ecological adaptation have also been found between the western and eastern Baltic cod stocks [54].

Ecotypes are not specific to cod but have been reported in several marine and freshwater fish species. The European anchovy (*Engraulis encrasicolus*) is a widely distributed marine fish with genetic and phenotypic differentiation between coastal and offshore populations of the species consistently being found through its range ([55] and references therein). The most striking ecotype examples in fish are found in freshwater, where substantial phenotypic and genetic divergence has taken place since the end of the last glacial period. Perhaps the clearest example of ecotype development is found in Arctic char (*Salvelinus alpinus*) in Þingvallavatn (Iceland), where significantly divergent ecotypes have developed rapidly in the last 10,000 years, offering an opportunity to study speciation [5,56–58]. While the clear morphological differences between the Arctic char ecotypes make them readily distinguishable, the identification of ecotypes in other species can be more complicated, as is the case for cod in the present study.

In summary, this study supports previous reports of discrimination of Icelandic cod ecotypes on the basis of otolith shape [12], but with important caveats that may limit broad applicability. The otolith-based method appears to be sensitive to both the broad habitat range of the intermediate *Pan* I$^{AB}$ genotype and a strong effect of growth rate on shape, which limits discriminatory power when there are substantial temporal and spatial variations in growth rate masking ecotype-related growth rate differences.

## Supporting information

**S1 Fig. Reconstruction of otolith shape using both Fourier and Wavelet descriptors.**
(DOCX)

**S1 Table. Parameter estimates for Table 5A.**
(XLSX)

**S2 Table. Parameter estimates for Table 5B.**
(XLSX)

**S1 Data.**
(XLSX)

## Acknowledgments

Staff of the Marine and Freshwater Research Institute (MFRI) collected and archived the samples used in the study, as well as supporting the current research with equipment, facilities and a helping hand. Special thanks to Sigrún Jóhannsdóttir for her help.

Statistical support was given by Denis Warshan, Bjarki Elvarsson and Sölvi Rögnvaldsson.

## Author Contributions

**Conceptualization:** Einar Pétur Jónsson, Steven E. Campana.

**Methodology:** Einar Pétur Jónsson, Steven E. Campana, Hlynur Bárðarson.

**Resources:** Klara B. Jakobsdóttir.

**Validation:** Jón Sólmundsson, Klara B. Jakobsdóttir, Hlynur Bárðarson.

**Visualization:** Einar Pétur Jónsson.

**Writing – original draft:** Einar Pétur Jónsson.

**Writing – review & editing:** Steven E. Campana, Jón Sólmundsson, Klara B. Jakobsdóttir.

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
