## [Decision Letter · Decision Letter 0]

9 Apr 2021

PONE-D-21-04437

Otolith-based discrimination of cod ecotypes and the effect of growth rate.

PLOS ONE

Dear Dr. Jónsson,

Thank you for submitting your manuscript to PLOS ONE. After careful consideration, we feel that it has merit but does not fully meet PLOS ONE’s publication criteria as it currently stands. Therefore, we invite you to submit a revised version of the manuscript that addresses the points raised during the review process. Three reviewers have evaluated your manuscript. They would like to see your work published, but Rev2 and Rev3 point to what they consider quite severe weaknesses. Especially these issues, but also the other points brought up by the reviewers must be addressed. If you disagree with any comment or requirement and choose not to change your ms in accordance, you must argue well for your case.

We look forward to receiving your revised manuscript.

Kind regards,

Geir Ottersen

Academic Editor

PLOS ONE

Journal Requirements:

We noted in your submission details that a portion of your manuscript may have been presented or published elsewhere.

„Yes. The genetic information used in the manuscript was previously published by one of the authors (Jakobsdóttir et al., 2011).

This data, which is only a part of the data underlying the current study, is used as a proxy for the ecotype identity here and used in a different manner than in Jakobsdóttir's study.”

Please clarify whether this publication was peer-reviewed and formally published. If this work was previously peer-reviewed and published, in the cover letter please provide the reason that this work does not constitute dual publication and should be included in the current manuscript.

We note that Figure 1 in your submission contain map images which may be copyrighted. All PLOS content is published under the Creative Commons Attribution License (CC BY 4.0), which means that the manuscript, images, and Supporting Information files will be freely available online, and any third party is permitted to access, download, copy, distribute, and use these materials in any way, even commercially, with proper attribution. For these reasons, we cannot publish previously copyrighted maps or satellite images created using proprietary data, such as Google software (Google Maps, Street View, and Earth). For more information, see our copyright guidelines: http://journals.plos.org/plosone/s/licenses-and-copyright.

3a, You may seek permission from the original copyright holder of Figure 1 to publish the content specifically under the CC BY 4.0 license. 

3b, If you are unable to obtain permission from the original copyright holder to publish these figures under the CC BY 4.0 license or if the copyright holder’s requirements are incompatible with the CC BY 4.0 license, please either i) remove the figure or ii) supply a replacement figure that complies with the CC BY 4.0 license. Please check copyright information on all replacement figures and update the figure caption with source information. If applicable, please specify in the figure caption text when a figure is similar but not identical to the original image and is therefore for illustrative purposes only.

Reviewers' comments:

Reviewer's Responses to Questions

**Comments to the Author**

1. Is the manuscript technically sound, and do the data support the conclusions?

Reviewer #1: Yes

Reviewer #2: Partly

Reviewer #3: Partly

2. Has the statistical analysis been performed appropriately and rigorously? 

Reviewer #1: Yes

Reviewer #2: Yes

Reviewer #3: Yes

3. Have the authors made all data underlying the findings in their manuscript fully available?

Reviewer #1: Yes

Reviewer #2: Yes

Reviewer #3: Yes

4. Is the manuscript presented in an intelligible fashion and written in standard English?

Reviewer #1: Yes

Reviewer #2: Yes

Reviewer #3: Yes

5. Review Comments to the Author

Reviewer #1: General comments:

This is a well-written MS on a thoroughly considered study on the discriminatory power of otolith shape analysis for Icelandic cod (ecotypes). The paper especially explores the effect of growth rate, which was found to drive differences in otolith shapes in previous studies, on the differentiation between ecotypes. This study uses genetically-typed material to contrast the morphometric results to. Overall, it adds to the current knowledge on the separation of Icelandic cod groups, which could be used for improving fisheries management of local cod stocks, also in other (similar) areas such as Norway, Greenland or Faroe Islands.

All text sections are appropriate in length and well-supported by tables and figures. The choice and description of material & methods, and the results sections are clear and easy to follow. The statistical analysis is sound and appropriate for the methods used and for supporting the argumentation. The introduction and discussion make reference to the relevant literature. The language is good and concise.

Editorial comments:

The numbering of references should be checked, e.g. line 70: should be references 19, 21 and 22?; line 76: should be refs 21, 23?; line 149: refs 12, 18?; line 145: ref 34?; line 373: ref 46?; line 374: ref 47?; line 388-404: shift in numbering by -1?

References #42, 45 and 46 do not appear to be cited in the text.

Ref 26 is published in vol. 77(3), 1043–1054. doi:10.1093/icesjms/fsz259

Reviewer #2: This manuscript describes the effect of variability in growth rate on the morphology of cod otoliths and the ability to use otolith morphology to discriminate between ecotypes within the Icelandic cod stock. While the methodology and results are sound, I have a major concern about the validity of the primary underlying assumption made by the authors, namely, that the Pan-I genotype is by itself diagnostic of ecotype in an individual cod. To the best of my knowledge, this is not supported by existing data which has found that all three genotypes of the Pan-I locus are present in both the coastal and frontal ecotypes. It is true that the frequency of the three genotypes (and their alleles) do vary between ecotypes at a group/population level. However, taking a single fish, genotyping it, and definitively assigning it to the coastal or frontal ecotype has not been proven to be a valid approach. There are habitat differences associated with the different genotypes of Pan-I where the relative frequency of the A and B alleles varies along a depth gradient, but this was independent of any of the other behaviors, morphological differences, physiological differences that distinguish the coastal and frontal ecotypes. Therefore, because this underlying assumption is not valid, the conclusions that the authors draw from their results are questionable as it relates to ecotype and it undermines that main objective of the manuscript. However, the authors have an impressive dataset and I can think of a couple of ways this concern could potentially be addressed. The work of Pardoe et al. (2009), Pardoe and Marteinsdottir (2009), Grabowski et al. (2011), McAdam et al. (2012), and Solmundsson et al. (2015) could be used to justify the classification individuals captured together as the same ecotype based on allele frequency and potentially the depth of the net set(s). This would probably require reconsidering how the spatial component of the existing analyses is handled as it may introduce considerable autocorrelation as currently constructed. The authors could also focus on differences in the morphology of the otoliths of individuals with different Pan-I genotypes and how growth rate influenced and draw more general conclusions on how these results may be related to ecotypes in their discussion.

Minor editorial comments follow:

L4-5: The title does not really encompass what the authors tested. In particular, the second part of the title is not very clear. If the authors are examining the effect of growth rate on their ability to discriminate between cod ecotypes using otoliths, as stated in their objectives, why not explicitly say so, e.g., “The effect of growth rate on otolith-based discrimination of cod ecotypes” or “Variability in growth rate masks differences in otolith morphology between cod ecotypes” or similar?

L28: Change to 58-year sampling period

L128: Report the frequency that this was necessary.

L134-135: Is there any evidence that the Feret diameter measures are not sensitive to this sort of otolith reconstruction, if so, please provide citations? Please report the number of times that this was necessary.

L158-160: It is not clear whether the fish used in this analysis had been previously genotyped by Jakobsdottir et al. or if they had been genotyped by the authors following the approach described in that paper. If the latter, the procedure followed should be either briefly summarized here (2-3 sentences) or detailed in supplementary materials.

L191-193: This statement should be supported with citations of the relevant literature.

Table5a: I am not aware of a situation in which P can be < 0.

L306-321: The authors make an implicit assumption that is hinted at in the introduction and becomes clearer here. As discussed above, the underlying assumption that the Pan-I genotype is by itself diagnostic of ecotype in an individual fish is not supported by existing data.

L340-342: Are the authors proposing that cod are segregated by strictly by Pan-I genotype?

L362-364: It is not clear how the authors determined the ecotype of individual cod based only on genotype data.

L407-408: The data and analysis presented do not really seem to support anything related to distinguishing ecotypes, only genotypes.

L409-411: This sentence is not very clear. How is there a broad distribution of the Pan-I AB genotype…it is a categorical state.

L437: The formatting of the references section is very inconsistent.

Figure captions are missing important and relevant information, e.g., common and scientific name of the study species, range of sampling years, etc.

Reviewer #3: General comments

The present paper by Jónsson et al investigates the effect of growth rate on the otolith shape of three ecotypes of cod. Earlier work has suggested that the cod can be successfully assigned to ecotype using otolith shape. However, shape is also affected by growth rate, which may weaken the relationship between ecotype and otolith shape. The study is therefore timely and relevant. The present study uses a 50+ year time series of otoliths collected at three locations and uses Pan I genotypes to discriminate between ecotypes. The shape and growth data appear valid. However, I have some concerns regarding the experimental design that I find could be improved.

The rationale behind using a long time-series to investigate the relationship is not clearly described in the paper and it comes at the expense of a complex and unbalanced design, which may render the analyses less straightforward than those obtained with a simpler design. Of course, the time-series allows cohort effects to be examined, but a shorter and more balanced sampling scheme may have allowed this too. The relative contributions of growth rate, cohort and location to shape variability is tested. However, these three factors may also be correlated.

Moreover, unquantified variability is introduced by the use of different capture techniques that may target different components of the population. A gear such as the longline, which is only used at one location and one year, will target fish actively feeding, whereas an active gear such as bottom trawl is less selective. The data show that the fish caught by long-line are very slow growing, and I guess these fish will have had a major impact on the relationship between growth and otolith shape. However, this could also be due to cohort and location effects, but this cannot be tested with the present design as there is only one sample obtained with long-lines. The present design may be improved by only selecting fish sampled by gill nets. It will shorten the time-series, but may make the test more robust. I would like the authors to reconsider, or at least argue in favor of, this part of their experimental design.

In summary, I find the study interesting and with potential. However, before I can recommend the paper for publication, I would like the authors to consider whether their experimental design can be improved to yield more robust test of the contribution of cohort, location and growth rate on otolith shape. Moreover, the specific comments below should be addressed.

Specific comments

L31: I would not call this high classification success given that the analysis only operates with two groups. I would call it “moderate” at best.

L68: I would write ”potential” instead of power. The otolith shape analysis is not yielding robust results in all cases, - as is the case in the present study.

L80ff: Long and convoluted sentence. I would suggest splitting it up into two sentences.

L102: Why are the long time series a requirement?

L113: Are there differences in age distribution (# cohorts) between the different ecotypes. Age differences may be important as otoliths may change shape as the fish age. Are age related tendencies removed during detrending? Also, are the catches for a given year consisting of catches from one fishing event (e.g. one particular gill net set) or do they consist of catches from several boats?

L176: What does the colors indicate? Ecotype?

L189: In line 91, it is written that the relative contribution of cohort, location and growth rate to shape variation is evaluated, but here the shape descriptors are evaluated against cohort, location and Pan I, whereas Growth rate is modelled based on ecotypes, cohort and locations. Please clarify.

L225: Yes, but reducing the number of groups would increase classifications success, - even if the fish were randomly distributed (from 33% to 50%)

L254: Does this suggest that the (weak) ability to discriminate between ecotypes is driven primarily by differences in growth? When growth is accounted for there is no effect of ecotype?

L274: The meaning of filled and open symbols is unclear. Does different symbols indicate significant differences or is it only filled symbols that are different from open symbols? Please clarify or use another system.

L318: If the location specific catches in a given year is based on one trawlhaul, one gill net set or one long-line set then there should be plenty of opportunity for random variation in the proportions, - especially if there is segregation in habitat among ecotypes. A small change in location or changes in the environmental conditions (frontal zone location etc) may led to different ecotypes being caught at the same general location during different years. To me that would be a much more likely explanation than large inter-annual variations in recruitment success of the different ecotypes.

L329: This should probably be the conclusion.

L333: Yes but this is for two groups instead of three.

L342: Yes, this is an important point. Maybe for a given sampling year and location classification success can be much higher, but in a temporally and spatially complex setting such as this, the variability in environmental factors makes otolith shape very variable.

L392: the general discussion on ecotypes should be reduced or cut out. It has been sufficiently dealt with in the introduction.

6. PLOS authors have the option to publish the peer review history of their article (what does this mean?). If published, this will include your full peer review and any attached files.

Reviewer #1: No

Reviewer #2: No

Reviewer #3: No

---

## [Author Response · Author response to Decision Letter 0]

25 Aug 2021

All of the reviewers provided very constructive and diplomatically-worded comments on the manuscript. They were much appreciated. Our responses to each of the comments is either shown below, implemented directly into the text, or both.

Editor: 

We noted in your submission details that a portion of your manuscript may have been presented or published elsewhere.

The material in this research article manuscript focuses on otolith use for identifying cod ecotype, none of which has been previously published. The one component that has been previously published is the genetic information on the Pantophysin I locus, which was used in our study only as a proxy for ecotype identity. There was no other focus on the genetics in our study, and the origin of the genetics data is clearly stated and cited in the manuscript. The original report on the genetics (Jakobsdóttir et al., 2011 - Historical changes in genotypic frequencies at the Pantophysin I locus in Atlantic cod (Gadus morhua) in Icelandic waters: evidence of fisheries-induced selection?) used it to examine historical trends in the genetic composition of the Icelandic cod stock, as evidence for changes in selection regimes and fishing pressure on the stock. The research objectives and content clearly differ between our study and that of Jakobsdottir et al. (2011), have very little overlap, and certainly cannot be classified as a dual publication. 

We note that Figure 1 in your submission contain map images which may be copyrighted. 

The map showing Iceland in Figure 1 is not copyrighted. Data behind the map were obtained from an open source R package (geo) 

https://github.com/Hafro/geo

Hoskuldur Bjornsson, Sigurdur Thor Jonsson, Arni Magnusson and Bjarki Thor Elvarsson (2018). geo: Draw and Annotate Maps, Especially Charts of the North Atlantic. R package version 1.5-0.

The map was plotted using the open source R package ggplot2:

H. Wickham. ggplot2: Elegant Graphics for Data Analysis. Springer-Verlag New York, 2016. 

Reviewer #1: General comments:

This is a well-written MS on a thoroughly considered study on the discriminatory power of otolith shape analysis for Icelandic cod (ecotypes). The paper especially explores the effect of growth rate, which was found to drive differences in otolith shapes in previous studies, on the differentiation between ecotypes. This study uses genetically-typed material to contrast the morphometric results to. Overall, it adds to the current knowledge on the separation of Icelandic cod groups, which could be used for improving fisheries management of local cod stocks, also in other (similar) areas such as Norway, Greenland or Faroe Islands.

All text sections are appropriate in length and well-supported by tables and figures. The choice and description of material & methods, and the results sections are clear and easy to follow. The statistical analysis is sound and appropriate for the methods used and for supporting the argumentation. The introduction and discussion make reference to the relevant literature. The language is good and concise.

Editorial comments:

The numbering of references should be checked, e.g. line 70: should be references 19, 21 and 22?; line 76: should be refs 21, 23?; line 149: refs 12, 18?; line 145: ref 34?; line 373: ref 46?; line 374: ref 47?; line 388-404: shift in numbering by -1? Done

References #42, 45 and 46 do not appear to be cited in the text. Done

Ref 26 is published in vol. 77(3), 1043–1054. doi:10.1093/icesjms/fsz259 Done

Reviewer #2:

 This manuscript describes the effect of variability in growth rate on the morphology of cod otoliths and the ability to use otolith morphology to discriminate between ecotypes within the Icelandic cod stock. While the methodology and results are sound, I have a major concern about the validity of the primary underlying assumption made by the authors, namely, that the Pan-I genotype is by itself diagnostic of ecotype in an individual cod. To the best of my knowledge, this is not supported by existing data which has found that all three genotypes of the Pan-I locus are present in both the coastal and frontal ecotypes. It is true that the frequency of the three genotypes (and their alleles) do vary between ecotypes at a group/population level. However, taking a single fish, genotyping it, and definitively assigning it to the coastal or frontal ecotype has not been proven to be a valid approach. There are habitat differences associated with the different genotypes of Pan-I where the relative frequency of the A and B alleles varies along a depth gradient, but this was independent of any of the other behaviors, morphological differences, physiological differences that distinguish the coastal and frontal ecotypes. Therefore, because this underlying assumption is not valid, the conclusions that the authors draw from their results are questionable as it relates to ecotype and it undermines that main objective of the manuscript. However, the authors have an impressive dataset and I can think of a couple of ways this concern could potentially be addressed. The work of Pardoe et al. (2009), Pardoe and Marteinsdottir (2009), Grabowski et al. (2011), McAdam et al. (2012), and Solmundsson et al. (2015) could be used to justify the classification individuals captured together as the same ecotype based on allele frequency and potentially the depth of the net set(s). This would probably require reconsidering how the spatial component of the existing analyses is handled as it may introduce considerable autocorrelation as currently constructed. The authors could also focus on differences in the morphology of the otoliths of individuals with different Pan-I genotypes and how growth rate influenced and draw more general conclusions on how these results may be related to ecotypes in their discussion.

Regarding your concern over the assignment of individual cod to either frontal or coastal based on their Pan I genotype on the individual level:

The accuracy of the relationship between ecotypes and the Pan I locus has now been made clearer in the introduction, as well as adding discussion about the level of error involved. It is acknowledged that the method involves a level of error, but it is however a convenient tool when no other ecotype approximation is at hand for such a long time series. We hope the increased clarity around the Pan I–ecotype relationship addresses your concern, but we have also tuned down the talk of ecotype where it is not appropriate and followed your advise by focusing more on the differences between genotypes and then moving onto the general ecotype connection in the discussion. 

Included is a graph from Hlynur Bardarson‘s unpublished paper which shows the percentages cited in the recently added discussion about the error involved in the Pan I-ecotype relationship. To clarify an error in this figure legend: AA is darkest, BB is whitest, with AB in between. 

Minor editorial comments follow:

L4-5: The title does not really encompass what the authors tested. In particular, the second part of the title is not very clear. If the authors are examining the effect of growth rate on their ability to discriminate between cod ecotypes using otoliths, as stated in their objectives, why not explicitly say so, e.g., “The effect of growth rate on otolith-based discrimination of cod ecotypes” or “Variability in growth rate masks differences in otolith morphology between cod ecotypes” or similar? Done

L28: Change to 58-year sampling period Done

L128: Report the frequency that this was necessary. Done

L134-135: Is there any evidence that the Feret diameter measures are not sensitive to this sort of otolith reconstruction, if so, please provide citations? Please report the number of times that this was necessary. Done

L158-160: It is not clear whether the fish used in this analysis had been previously genotyped by Jakobsdottir et al. or if they had been genotyped by the authors following the approach described in that paper. If the latter, the procedure followed should be either briefly summarized here (2-3 sentences) or detailed in supplementary materials. Done

L191-193: This statement should be supported with citations of the relevant literature. Done

Table5a: I am not aware of a situation in which P can be < 0. Done

L306-321: The authors make an implicit assumption that is hinted at in the introduction and becomes clearer here. As discussed above, the underlying assumption that the Pan-I genotype is by itself diagnostic of ecotype in an individual fish is not supported by existing data. The paragraph in question only focuses on the observations in a genotype context. More support, explanation and reasoning for the use of the Pan as a proxy for ecotype is provided elsewhere.

L340-342: Are the authors proposing that cod are segregated by strictly by Pan-I genotype? The paragraph in question only focuses on the observations in a genotype context. More support, explanation and reasoning for the use of the Pan as a proxy for ecotype is provided elsewhere. 

L362-364: It is not clear how the authors determined the ecotype of individual cod based only on genotype data. No, only that based on the result that the AB shape ranges from being similar to AA to being similar to BB shape, it seems likely that there are individuals of the AB genotype that encounter AA-like conditions and others that encounter BB-like conditions. Therefore they present greater otolith shape variability. For observations of the variability in behaviour of the AB genotype, see for example Pampoulie et al. (2008).

L407-408: The data and analysis presented do not really seem to support anything related to distinguishing ecotypes, only genotypes. Agreed. The current study supports the previous findings relating directly to behavioural ecotypes (DST profiles), but here using the Pan as a proxy. As a relationship has been shown between Pan and behavioural ecotypes, the support is rendered through that connection, without having DST profiles at hand for these archived samples. 

I hope this short conclusion may stand as it is because it refers to the previous study and also makes the connection to ecotypes - a central connection in the paper which has now been backed up and discussed. 

L409-411: This sentence is not very clear. How is there a broad distribution of the Pan-I AB genotype…it is a categorical state. Done

L437: The formatting of the references section is very inconsistent. Done

Figure captions are missing important and relevant information, e.g., common and scientific name of the study species, range of sampling years, etc. Done

Reviewer #3: General comments

The present paper by Jónsson et al investigates the effect of growth rate on the otolith shape of three ecotypes of cod. Earlier work has suggested that the cod can be successfully assigned to ecotype using otolith shape. However, shape is also affected by growth rate, which may weaken the relationship between ecotype and otolith shape. The study is therefore timely and relevant. The present study uses a 50+ year time series of otoliths collected at three locations and uses Pan I genotypes to discriminate between ecotypes. The shape and growth data appear valid. However, I have some concerns regarding the experimental design that I find could be improved.

The rationale behind using a long time-series to investigate the relationship is not clearly described in the paper and it comes at the expense of a complex and unbalanced design, which may render the analyses less straightforward than those obtained with a simpler design. Of course, the time-series allows cohort effects to be examined, but a shorter and more balanced sampling scheme may have allowed this too. The relative contributions of growth rate, cohort and location to shape variability is tested. However, these three factors may also be correlated. 

The samples used in the study were archived otoliths that had been genotyped for the Pan I locus, offering unique data that was available to be analysed. The long time-series not only includes the variability in growth rate neccessary to answer the principal research question, but also gives a long-term perspective on ecological processes, thus filtering out short-term anomalies and processes. The complex nature of the experimental design has now been clearly acknowledged and the value of the data emphasized in the text. 

Moreover, unquantified variability is introduced by the use of different capture techniques that may target different components of the population. A gear such as the longline, which is only used at one location and one year, will target fish actively feeding, whereas an active gear such as bottom trawl is less selective. The data show that the fish caught by long-line are very slow growing, and I guess these fish will have had a major impact on the relationship between growth and otolith shape. However, this could also be due to cohort and location effects, but this cannot be tested with the present design as there is only one sample obtained with long-lines. The present design may be improved by only selecting fish sampled by gill nets. It will shorten the time-series, but may make the test more robust. I would like the authors to reconsider, or at least argue in favor of, this part of their experimental design.

This is undeniably a possible source of unquantifiable variability in the data. As there is no year-location combination that has more than one type of gear, there cannot be made a comparison of the selectivity of different gears. All the gear types are however catching fully recruited sizes of cod for all ecotypes and the fish are almost entirely either spawning or mature for all the catches, regardless of gear type, with no clear difference between catches. It is therefore likely that the majority of these fish caught on spawning grounds are adult spawning fish. Cutting out some of the sample years due to different gear would be a costly reduction in data and its span which we feel would greatly alter the nature of this paper. 

This has now been commented on in the manuscript and we hope you find the matter sufficiently dealt with. 

In summary, I find the study interesting and with potential. However, before I can recommend the paper for publication, I would like the authors to consider whether their experimental design can be improved to yield more robust test of the contribution of cohort, location and growth rate on otolith shape. Moreover, the specific comments below should be addressed.

Specific comments

L31: I would not call this high classification success given that the analysis only operates with two groups. I would call it “moderate” at best. Done

L68: I would write ”potential” instead of power. The otolith shape analysis is not yielding robust results in all cases, - as is the case in the present study. Done

L80ff: Long and convoluted sentence. I would suggest splitting it up into two sentences. Done

L102: Why are the long time series a requirement? Done

L113: Are there differences in age distribution (# cohorts) between the different ecotypes. Age differences may be important as otoliths may change shape as the fish age. Are age related tendencies removed during detrending? Also, are the catches for a given year consisting of catches from one fishing event (e.g. one particular gill net set) or do they consist of catches from several boats?

The age ranges are nearly identical across ecotypes and locations and all of the fish are mature. The relevance of age is to cohort and cohort effect are dealt with in the analysis. Detrending length effects is the standard approach (i.e. Campana and Casselman, 1990). We are then testing for growth differences, which is length at age, so age is the effect that is really being tested. Therefore the detrending of age-related tendencies is not applicable. See table below.

The catches each year consist in some cases of several fishing events (years 1957, 1975, 1996 and 2000), while in other years it‘s a single event. 

L176: What does the colors indicate? Ecotype? Done

L189: In line 91, it is written that the relative contribution of cohort, location and growth rate to shape variation is evaluated, but here the shape descriptors are evaluated against cohort, location and Pan I, whereas Growth rate is modelled based on ecotypes, cohort and locations. Please clarify. Done

L225: Yes, but reducing the number of groups would increase classifications success, - even if the fish were randomly distributed (from 33% to 50%). This is now discussed in discussion.

L254: Does this suggest that the (weak) ability to discriminate between ecotypes is driven primarily by differences in growth? When growth is accounted for there is no effect of ecotype?

To a certain extent it does, yes. We suspect that otolith shape is a useful discriminant only when the growth rate differences are strong enough between groups. 

This analysis in particular indicates that when the effect of growth rate on otolith shape is accounted for, the genotype effect is insignificant. 

L274: The meaning of filled and open symbols is unclear. Does different symbols indicate significant differences or is it only filled symbols that are different from open symbols? Please clarify or use another system. Clarified

L318: If the location specific catches in a given year is based on one trawlhaul, one gill net set or one long-line set then there should be plenty of opportunity for random variation in the proportions, - especially if there is segregation in habitat among ecotypes. A small change in location or changes in the environmental conditions (frontal zone location etc) may led to different ecotypes being caught at the same general location during different years. To me that would be a much more likely explanation than large inter-annual variations in recruitment success of the different ecotypes. Wording revised. 

L329: This should probably be the conclusion. If you refer to the otolith shape differences not being strong enough to allow for successful discrimination, then this information is included in the conclusion paragraph – the last paragraph in the paper. There is also mention of the discrimination trouble caused by the AB genotype. 

L333: Yes but this is for two groups instead of three. Now discussed.

L342: Yes, this is an important point. Maybe for a given sampling year and location classification success can be much higher, but in a temporally and spatially complex setting such as this, the variability in environmental factors makes otolith shape very variable. Agreed.

L392: the general discussion on ecotypes should be reduced or cut out. It has been sufficiently dealt with in the introduction. Done

location genotype mean age age range

all pooled AA 7.6 6 to 13

all pooled AB 8.3 6 to 14

all pooled BB 9.6 6 to 14

BRE AA 7.7 6 to 11

BRE AB 8.3 6 to 14

BRE BB 9.5 6 to 14

FAX AA 7.4 6 to 13

FAX AB 7.7 6 to 11

FAX BB 9 7 to 11

SW AA 7.6 6 to 14

SW AB 8.7 6 to 14

SW BB 9.7 6 to 14

---

## [Editor Report · Decision Letter 1]

10 Sep 2021

The effect of growth rate on otolith-based discrimination of cod (Gadus morhua) ecotypes

PONE-D-21-04437R1

Dear Dr. Jónsson,

We’re pleased to inform you that your manuscript has been judged scientifically suitable for publication and will be formally accepted for publication once it meets all outstanding technical requirements.

Kind regards,

Geir Ottersen

Academic Editor

PLOS ONE

Additional Editor Comments (optional):

I find you have done a good and thorough job in revising according to the reviewers requests. I am pleased that you have documented the revison well in the Response to Revewers letter, the answer to PLOS (Lilla Petho) and a "track changes" version of your ms. Good that you clearly have resolved the suggested potential issues relating to the paper by Jakobsdottir et al. in Ecol. Applications.

---

## [Editor Report · Acceptance letter]

20 Sep 2021

PONE-D-21-04437R1 

The effect of growth rate on otolith-based discrimination of cod (*Gadus morhua*) ecotypes 

Dear Dr. Jónsson:

I'm pleased to inform you that your manuscript has been deemed suitable for publication in PLOS ONE. Congratulations! Your manuscript is now with our production department. 

Kind regards, 

on behalf of

Dr. Geir Ottersen 

Academic Editor

PLOS ONE